# Dapagliflozin Prevents High-Glucose-Induced Cellular Senescence in Renal Tubular Epithelial Cells

**DOI:** 10.3390/ijms232416107

**Published:** 2022-12-17

**Authors:** Theodoros Eleftheriadis, Georgios Pissas, Georgios Filippidis, Maria Efthymiadi, Vassilios Liakopoulos, Ioannis Stefanidis

**Affiliations:** Department of Nephrology, Faculty of Medicine, University of Thessaly, Biopolis, Mezourlo Hill, 41110 Larissa, Greece

**Keywords:** dapagliflozin, senescence, oxidative stress, DNA damage, diabetic nephropathy

## Abstract

Gliflozins are a new class of antidiabetic drugs with renoprotective properties. In cultures of primary human renal tubular epithelial cells (RPTECs) subjected to high-glucose conditions in the presence or absence of dapagliflozin, we evaluated cellular senescence pathways. High glucose increased sodium–glucose cotransporter-2 (SGLT-2) expression and glucose consumption, enhancing reactive oxygen species production. The latter induced DNA damage, ataxia telangiectasia mutated kinase (ATM), and p53 phosphorylation. Stabilized p53 increased the cell cycle inhibitor p21, resulting in cell cycle arrest and increasing the cellular senescence marker beta-galactosidase (GLB-1). RPTECs under high glucose acquired a senescence-associated secretory phenotype, which was detected by the production of IL-1β, IL-8, and TGF-β1. By decreasing SGLT-2 expression and glucose consumption, dapagliflozin inhibited the above pathway and prevented RPTEC senescence. In addition, dapagliflozin reduced the cell cycle inhibitor p16 independently of the glucose conditions. Neither glucose concentration nor dapagliflozin affected the epithelial-to-mesenchymal transition when assessed with α-smooth muscle actin (α-SMA). Thus, high glucose induces p21-dependent RPTEC senescence, whereas dapagliflozin prevents it. Since cellular senescence contributes to the pathogenesis of diabetic nephropathy, delineating the related molecular mechanisms and the effects of the widely used gliflozins on them is of particular interest and may lead to novel therapeutic approaches.

## 1. Introduction

Diabetic nephropathy (DN) is the leading cause of end-stage kidney disease, and the need for new medications to modify its development and progression is urgent. Recently, a new class of drugs, the sodium–glucose cotransporter 2 (SGLT-2) inhibitors gliflozins, have been proven protective against the progression of DN, as well as against other forms of chronic kidney disease [1,2].

Gliflozins reduce glucose and sodium reabsorption by renal proximal tubular epithelial cells (RPTECs), trigger tubuloglomerular feedback mechanisms, induce afferent arteriolar vasoconstriction, and eventually decrease intraglomerular pressure [3]. Although the renoprotective effects of gliflozins have been attributed to such hemodynamic parameters, accumulated data suggest that the amelioration of direct glucotoxicity in RPTECs plays a significant role [4,5,6,7].

Filtered by the glomeruli, glucose enters RPTECs through SGLT-2. Filtered glucose is the primary source of glucose for RPTECs, since almost 90% of glucose is absorbed by these cells, and no glucose appears in the urine unless blood glucose exceeds 180 mg/mL [8]. Thus, the glucose load is high enough for RPTECs even in the absence of diabetes mellitus. The latter becomes worse in the case of chronic kidney disease as, although the total glomerular filtration rate (GFR) decreases, single-nephron GFR may increase, and more glucose is filtered per nephron [9,10]. Thus, in chronic kidney disease, the glucose load for RPTECs may increase, and in such cases, gliflozins may exert a renoprotective role even in the absence of diabetes mellitus.

RPTEC senescence characterizes DN and contributes to its pathogenesis and progression [11,12,13]. Senescent cells enter a phase of permanent cell cycle arrest. In addition, they acquire a senescence-associated secretory phenotype (SASP), secreting various proinflammatory and profibrotic cytokines and contributing to the pathogenesis of DN [12,13]. Furthermore, permanent cell cycle arrest prevents dedifferentiation and proliferation of RPTECs in the case of acute kidney injury, thus decreasing the possibility of renal function recovery [13,14]. The latter may contribute to the susceptibility of patients with diabetes mellitus to acute kidney injury and to a worse prognosis after exposure to various factors, such as radiocontrast media [15].

In a previous study, cellular senescence was observed in the RPTECs of diabetic mice and in human RPTEC cultures, and SGLT-2 gene silencing was found to prevent cell senescence [16]. In this study, we evaluated the clinically relevant question of whether the already-in-clinical-use SGLT-2 inhibitor dapagliflozin prevents cell senescence in RPTECs exposed to high-glucose concentrations. The molecular pathways involved in high-glucose-induced RPTEC senescence and how they are affected by dapagliflozin were also assessed.

To avoid the effects of genetic modification, we did not use cell lines or immortalized cells, but primary human RPTECs. In addition, we used a dapagliflozin concentration that was close to the observed in a clinic. After the usual oral administration of 10 mg of dapagliflozin, its total serum maximum concentration (Cmax) was 160–170 ng/mL [17]. Gliflozins reach SGLT-2 from the luminal site of RPTECs [18,19]. In our experiments, we used the therapeutic concentration of 15 ng/mL, since 90% of the total circulating dapagliflozin is protein-bound and, consequently, unable to cross the glomerular filtration barrier and interact with SGLT-2 [17,19,20].

## 2. Results

### 2.1. High Glucose Increases Glucose Consumption and SGLT2 Expression, While Dapagliflozin Decreases Both

In RPTECs cultured under high-glucose conditions, glucose consumption was increased. Dapagliflozin reduced glucose consumption under both normal and high-glucose conditions. In the second case, it did so to a greater extent (Figure 1A).

In RPTECs cultured under high-glucose conditions, SGLT-2 expression was increased. Dapagliflozin reduced SGLT-2 expression under normal conditions and normalized it under high-glucose conditions (Figure 1B,C).

### 2.2. Dapagliflozin Decreases High-Glucose-Induced ROS Overproduction

In RPTECs, the high-glucose conditions increased the production of reactive oxygen species (ROS), whereas dapagliflozin decreased the high-glucose-induced ROS overproduction (Figure 2A).

ROS production was also assessed indirectly by using the level of 4-HNE-modified proteins. High-glucose conditions increased the level of 4-Hydroxynonenal (4-HNE)-modified protein, while dapagliflozin reduced the high-glucose-induced 4-HNE-modified protein level (Figure 2B,C).

### 2.3. Dapagliflozin Prevents High-Glucose-Induced DNA Damage

Under high-glucose conditions, ROS overproduction resulted in DNA damage, as was assessed by the level of phosphorylation at Ser139 histone H2AX (γ-H2AX). In RPTECs cultured under a high glucose concentration, dapagliflozin normalized the level of γ-H2AX (Figure 3A,D).

The high-glucose DNA damage elicited a DNA damage response (DDR). In RPTECs cultured under a high glucose concentration, the phosphorylated ataxia telangiectasia mutated kinase (p-ATM) to ATM ratio increased, whereas dapagliflozin normalized the above ratio (Figure 3A,B). The increase in p-ATM levels due to DNA damage led to an increase in the phosphorylation levels of tumor suppressor p53 (p53). Indeed, under high-glucose conditions, the p-p53 to p53 ratio increased, while dapagliflozin normalized the above ratio (Figure 3A,C).

### 2.4. Dapagliflozin Prevents High-Glucose-Induced p21 Upregulation and Decreases p16 Expression Irrespectively of Glucose Conditions

The phosphorylation and stabilization of p53 under high-glucose conditions upregulated the cell cycle repressor p21 Waf1/Cip1 (p21). In RPTECs cultured under a high glucose concentration, p21 expression was elevated, whereas dapagliflozin prevented p21 overexpression (Figure 4A,B).

High glucose did not affect the expression of the cell cycle repressor p16 INK4A (p16) in RPTECs. Dapagliflozin reduced the p16 levels under normal and high-glucose conditions (Figure 4A,C).

### 2.5. Dapagliflozin Decreases Markers of Cellular Senescence

High glucose decreased the level of the marker of proliferation Ki-67 (Ki-67) in RPTECs. Dapagliflozin normalized the high-glucose-induced Ki-67 downregulation (Figure 4A,D).

High-glucose conditions induced a substantial increase in the marker of cellular senescence, beta-galactosidase (GLB-1). On the other hand, dapagliflozin decreased GLB-1 in RPTECs under normal glucose conditions and normalized GLB-1 expression under high-glucose conditions (Figure 4A,E).

### 2.6. Dapagliflozin Ameliorates the High-Glucose-Induced SASP Phenotype

RPTECs cultured under a high glucose concentration acquired an SASP phenotype. High glucose triggered interleukin-1β (IL-1β) production by the cells, whereas dapagliflozin ameliorated high-glucose-induced IL-1β overproduction (Figure 5A).

The interleukin-8 (IL-8) concentration increased in the supernatants of RPTECs cultured under high-glucose conditions. Dapagliflozin decreased the high-glucose-induced IL-8 overproduction (Figure 5B).

RPTECs cultured under high-glucose conditions produced more transforming growth factor-β1 (TGF-β1). Dapagliflozin normalized the high-glucose-induced TGF-β1 overproduction (Figure 5C).

### 2.7. Neither High Glucose Nor Dapagliflozin Affects the Epithelial-to-Mesenchymal Transition

The epithelial-to-mesenchymal transition was assessed according to the α-smooth muscle actin (α-SMA) levels. High glucose did not alter the α-SMA expression in RPTECs. Under both normal and high-glucose conditions, dapagliflozin left the α-SMA levels unaffected (Figure 5D,E).

## 3. Discussion

Although gliflozins were developed as antidiabetic drugs, it was soon realized that they exerted nephroprotective effects [1,2]. Aside from their beneficial effects on glomerular hemodynamics [3], accumulated evidence suggests that by downregulating the tremendous glucose load of RPTECs, especially in the case of DM, gliflozins play a direct protective role in these cells [4,5,6,7]. For instance, a previous study showed that under high-glucose conditions, glucose influx into RPTECs increased, leading to ROS overproduction and metabolic alterations that diverted glucose metabolism towards four noxious pathways, the polyol pathway, the hexosamine pathway, the protein kinase C pathway, and the advanced glycation end-product pathway. Eventually, these pathways resulted in the production of proinflammatory and profibrotic cytokines and apoptosis in RPTECs [7].

In addition to RPTEC death, RPTEC senescence has been observed in DN and contributes to its pathogenesis and progression, since senescent cells acquire an SASP phenotype and excrete proinflammatory and profibrotic cytokines [11,12,13]. In addition, cell senescence limits the regenerative capacities of the kidneys, a fact that may be responsible for the vulnerability of patients with diabetes mellitus to acute kidney injury [14,15].

In this study, we evaluated the effects of high glucose concentrations in RPTEC cultures on the pathways that lead to cellular senescence and the impact of the clinically used SGLT-2 inhibitor dapagliflozin.

High glucose concentrations increased the influx of glucose into RPTECs, while dapagliflozin decreased the entry of glucose into the cells. Interestingly, high glucose also led to SGLT-2 overexpression. From a teleological point of view, high-glucose-induced SGLT-2 overexpression protects the organism from energy waste. Dapagliflozin reduced glucose consumption by RPTECs and also decreased SGLT-2 expression under both normal and high-glucose conditions. The molecular mechanism involved in the downregulation of SGLT-2 by dapagliflozin needs to be evaluated. However, this was also observed in previous studies with dapagliflozin or empagliflozin [7,21,22].

ROS production was evaluated directly, as well as indirectly, according to the level of 4-HNE-modified proteins. The increased influx of glucose into the RPTECSs under high-glucose conditions led to a burst of ROS production. Under high-glucose conditions, the initial ROS source was the mitochondria, but later, other sources were also involved, such as NADPH oxidase or xanthine oxidase [23,24,25]. By decreasing the influx of glucose into RPTECs, dapagliflozin significantly ameliorated the high-glucose-induced oxidative stress. Although dapagliflozin also decreased the influx of glucose into RPTECs cultured under a normal glucose concentration, the inhibitor did not exert any alterations in ROS production. The latter could be attributed to the smaller degree of the dapagliflozin-induced decrease in glucose consumption under normal glucose conditions compared to high-glucose conditions.

Oxidative stress affects DNA integrity and elicits the DNA damage response. Indeed, in RPTECs, the high-glucose-induced ROS overproduction caused DNA damage was assessed according to the levels of the modified histone γ-H2AX. The DNA damage response was also elicited because, under high glucose, increased levels of p-ATM were detected. ATM is a serine/threonine protein kinase that senses double-strand breaks in DNA. Then, it is autophosphorylated and activated [26]. Tumor suppressor p53 is an ATM target [26]. The p53 phosphorylation protects it from ubiquitination and degradation, thus allowing the transcription of p21, a well-known cell-cycle inhibitor [27], in addition to other p53 transcriptional targets [28]. In our experiments, high glucose increased the p21 levels, whereas dapagliflozin prevented high-glucose-induced upregulation of p21. Since p21 induces cell cycle arrest, it is considered a marker of cellular senescence [29]. The critical role of p21 in inducing RPTEC senescence under high glucose was also detected in a previous study in which p21 gene silencing prevented cellular senescence [16].

Another cell cycle inhibitor that is also considered a senescence marker is p16 [29,30]. Although, among various other factors, DNA damage is regarded as a p16 inducer [31], in our experimental model, the p16 levels remained unaffected in RPTECs cultured under high-glucose conditions. Thus, p16 may not play a significant role in high-glucose-induced cellular senescence. Interestingly, in a mouse model of DN, RPTEC senescence was revealed in immunohistochemistry, along with p21 upregulation, whereas p16 was not detected [16]. In our study, dapagliflozin decreased the p16 levels in RPTECs under normal and high-glucose conditions. The molecular mechanisms involved in the dapagliflozin-induced downregulation of p16 need to be evaluated. Another interesting question is that of whether the dapagliflozin-induced downregulation of p16 plays a permissive role in the protective effect of the drug under high-glucose conditions.

Cellular senescence is characterized by permanent cell cycle arrest. Accordingly, in our experimental model, Ki-67, a cell proliferation marker [32], decreased significantly under high-glucose conditions, whereas dapagliflozin prevented its downregulation. As regards GLB-1, the most commonly used marker of cellular senescence [29], high glucose almost doubled its level. Dapagliflozin normalized the high-glucose-induced upregulation of GLB-1. Interestingly, in RPTECs cultured under normal glucose conditions, dapagliflozin decreased the level of GLB-1 below the control levels, but did not affect Ki-67.

In addition to cell cycle arrest, senescent cells acquire an SASP phenotype and excrete proinflammatory and profibrotic cytokines, which, by acting in an autocrine and a paracrine manner, promote organ dysfunction [12,13]. We evaluated the production of three cytokines by RPTECs, and we detected that the IL-1β, IL-8, and TGF-β1 concentrations in cell culture supernatants increased significantly under high-glucose conditions. Dapagliflozin decreased the high-glucose-induced IL-1β and IL-8 production and normalized the level TGF-β1. In addition to inflammation, IL-1β can activate adjacent endothelial cells and aggravate diabetic microangiopathy and disease progression [33]. In addition, IL-1β can induce the endothelial-to-mesenchymal transition and accelerate fibrosis [34], the hallmark of DN [35,36,37]. IL-8 is a chemokine that attracts and activates neutrophils [38]. Peritubular neutrophil and macrophage infiltration was associated with a faster progression of DN [39]. In addition, there are data on the role of IL-8 in angiogenesis and the epithelial-to-mesenchymal transition [38], both of which also characterize DN [35,36,37]. Finally, TGF-β1 is considered the archetypical profibrotic cytokine. TGF-β1 promotes fibrosis by inducing regional transformation of fibroblasts into myofibroblasts and the epithelial-, endothelial-, pericyte-, or fibrocyte-to-mesenchymal transition [35,36,37].

Although high glucose induced a five-fold increase in TGF-β1, it did not cause an epithelial-to-mesenchymal transition according to the myofibroblast marker α-SMA [36,37]. Accordingly, dapagliflozin did not affect the α-SMA levels. Generally, there is no consensus about the role of the epithelial-to-mesenchymal transition in the pathogenesis of DN. Some studies suggest a significant role, whereas others suggest the opposite [36,37]. In our experimental model, high glucose did not seem to induce an epithelial-to-mesenchymal transition. However, the increased production of profibrotic cytokines by RPTECs is likely to contribute to renal fibrosis by affecting adjacent cell types.

A limitation of our study is its in vitro nature. However, the strict conditions of our experimental system allowed us to evaluate the effects of high glucose and dapagliflozin in isolated RPTECs, i.e., the renal cells that are directly affected by this medication. This approach excludes other confounding factors, such as the hemodynamic effects of gliflozins, and it helps to clarify the renoprotective mechanisms of this new class of drugs at the molecular level. Figure 6 depicts the results of our study.

## 4. Materials and Methods

### 4.1. Cell Culture Conditions

Primary human RPTCEs (ScienCell, Carlsbad, CA, USA) were used for our experiments. Low-glucose DMEM (5.5 mM D-glucose) (Thermo Fisher Scientific Inc., Rochford, IL, USA) or high-glucose DMEM (25 mM D-glucose) (Thermo Fisher Scientific Inc.) was used as a culture medium. The concentration of 5.5 mM D-glucose corresponded to the normal D-glucose concentration. To achieve equal osmolarity with the high-glucose culture medium (25 mM D-glucose), L-glucose (Sigma-Aldrich, St. Louis, MO, USA; Merck Millipore, Darmstadt, Germany) at a concentration of 19.5 mM was added to the cells that were cultured under a normal glucose concentration (5.5 mM D-glucose). Cells were cultured with or without 15 ng/mL dapagliflozin (Selleck Chemicals, Munich, Germany) for 24 h. RPTECs were cultured in 6-well plates (3 × 10^5^ cells) or 96-well plates (1 × 10^4^ cells) in a humidified atmosphere containing 5% CO_2_ and at 37 °C. Each experiment was repeated three times.

### 4.2. Assessment of Glucose Consumption and IL-1β, IL-8, and TGF-β1 Production

D-glucose consumption and the concentration of interleukin (IL)-1β, IL-8, and transforming growth factor (TGF)-β1 were measured in the supernatants of the RPTECs cultured in 6-well plates. An Element Blood Glucose Monitor (Element, Infopia, Titusville, FL, USA) was used to measure D-glucose. Glucose consumption was calculated by subtracting each measurement from the initial D-glucose concentration of the culture medium. IL-1β was measured with the Human IL-1β PLATINUM ELISA Kit (ca. no. BMS224/2, Bender MedSystems, Vienna, Austria) with a sensitivity of 0.3 pg/mL. The Human IL-8/NAP-1 ELISA Kit (Bender MedSystems) was used to assess IL-8. The detection range of this kit was 15.6–1000 pg/mL. TGF-β1 was measured with the Human TGF-beta-1 ELISA Kit (AssayPro, St. Charles, MO, USA). The detection range of this kit was 31–2000 pg/mL. The ELISA measurements were performed with the EnSpire Multimode Plate Reader (Perkin Elmer, Waltham, MA, USA). These experiments were repeated three times.

### 4.3. Assessment of Cellular ROS Production

To assess the reactive oxygen species (ROS) production, RPTECs were cultured in 96-well plates. After 24 h of cell culture, 5 μM of the fluorogenic probe CellROX Deep Red Reagent (Invitrogen, Life Technologies, Carlsbad, CA, USA) was used for 30 min at 37 °C. Then, the RPTECs were washed with phosphate buffer saline (Sigma-Aldrich; Merck Millipore), and the fluorescence signal intensity was assessed on an EnSpire Multimode Plate Reader (Perkin Elmer). These experiments were repeated three times.

### 4.4. Assessment of Cellular Proteins of Interest

To assess specific cellular protein levels, RPTECs were cultured in 6-well plates. For RPTEC lysis, the T-PER tissue protein extraction reagent (Thermo Fisher Scientific Inc., Waltham, MA, USA) supplemented with protease and phosphatase inhibitors (Sigma-Aldrich; Merck Millipore, and Roche Diagnostics, Indianapolis, IN, USA, respectively) was used. A Bradford assay (Sigma-Aldrich; Merck Millipore) was used to measure the protein concentration, and 10 μg of protein from each sample was electrophoresed in sodium dodecyl sulfate–polyacrylamide gel (4–12% Bis-Tris gels, Thermo Fisher Scientific Inc.) and transferred onto a polyvinylidene fluoride (PVDF) membrane (Thermo Fisher Scientific Inc.). Western blot bands were detected with the LumiSensor Plus Chemiluminescent HRP Substrate Kit (GenScript Corporation, Piscataway, NJ, USA). The Restore Western Blot Stripping Buffer (Thermo Fisher Scientific Inc.) was used to reprobe the PVDF membranes. Analysis of the Western blot bands was performed with the Image J software version 1.53f (National Institute of Health, Bethesda, MD, USA). These experiments were repeated three times.

Primary antibodies were applied for 16 h at 4 °C, while secondary antibodies were applied for 30 min at room temperature. The primary antibodies were specific for the following proteins: SGLT-2 (1:200, cat. No AGT-032, Alomone Labs, Jerusalem, Israel), 4-Hydroxynonenal (4-HNE, 1:500, cat. No ab46545 Abcam, Cambridge, UK), ataxia telangiectasia mutated kinase (ATM, 1:1000, cat. No 2873, Cell Signaling Technology, Danvers, MA, USA), phosphorylated at Ser1981 ATM (p-ATM, 1:1000, cat. No 5883, Cell Signaling Technology); tumor suppressor p53 (p53, 1:1000, cat. No 2524, Cell Signaling Technology), phosphorylated at Ser15 p53 (p-p53, 1:1000, cat. No 9284, Cell Signaling Technology), phosphorylated at Ser139 histone H2AX (γ-H2AX, 1:500, cat. No NB100-2280, Novus Biologicals, Abingdon, Oxon, UK), p21 Waf1/Cip1 (p21, 1:1000, cat. No 37543, Cell Signaling Technology), p16 INK4A (p16, 1:1000, cat. No 80772, Cell Signaling Technology), marker of proliferation Ki-67 (Ki-67, 1:1000, cat no NBP2-22112, Novus Biologicals), beta-galactosidase (GLB-1, 1:500, cat. No ab55176, Abcam), α-smooth muscle actin (α-SMA, 1:100, cat no. sc-130617; Santa Cruz Biotechnology, Dallas, TC, USA), and β-actin (1:2500, cat. no 4967, Cell Signaling Technology). The anti-rabbit IgG, HRP-linked antibody (1:1000, cat. no 7074, Cell Signaling Technology) and the anti-mouse IgG, HRP-linked antibody (1:1000, cat. no 7076, Cell Signaling Technology) were used as secondary antibodies.

### 4.5. Statistical Analysis

All statistical analyses were executed with IBM SPSS Statistics for Windows, Version 26 (IBM Corp., Armonk, NY, USA). A one-way analysis of variance (ANOVA) was used to compare means. Error bars represent the standard errors of the means, and *p*-values smaller than 0.05 were considered statistically significant.

## 5. Conclusions

In conclusion, by increasing glucose influx and ROS production, high glucose elicits the DNA damage response and, eventually, p21-dependent RPTEC senescence. Dapagliflozin decreases the influx of glucose into RPTECs and prevents high-glucose-induced cellular senescence. Since cellular senescence contributes to the pathogenesis of DN, delineating the related molecular mechanisms, as well as the effects that the widely used gliflozins have on them, is of particular interest and may lead to novel therapeutic approaches.

## Figures and Tables

**Figure 1 ijms-23-16107-f001:**
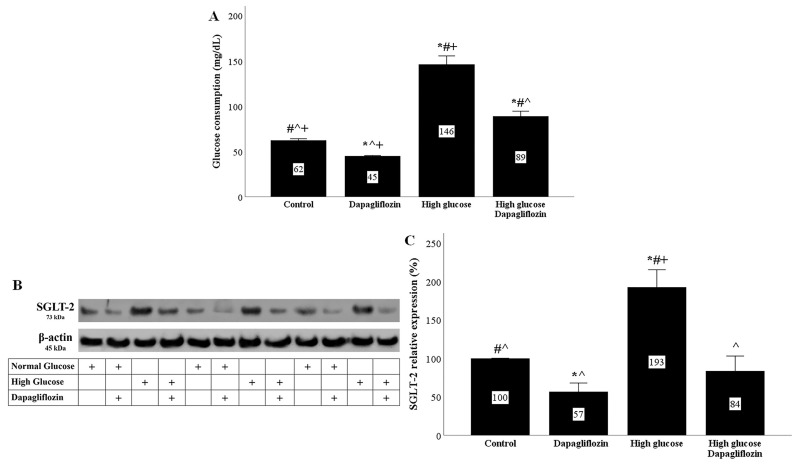
High glucose increased glucose consumption and SGLT2 expression, while dapagliflozin decreased both. In high-glucose conditions, glucose consumption was increased. Dapagliflozin reduced glucose consumption under normal and high-glucose conditions (**A**). High glucose upregulated SGLT2 expression. Dapagliflozin decreased SGLT-2 expression under normal and high-glucose conditions (**B**,**C**). * *p* < 0.05 vs. control; # *p* < 0.05 vs. RPTECs treated with 15 ng/mL dapagliflozin; ^ *p* < 0.05 vs. RPTECs in high glucose; + *p* < 0.05 vs. RPTECs in high glucose and treated with 15 ng/mL dapagliflozin. SGLT-2, sodium–glucose cotransporter-2.

**Figure 2 ijms-23-16107-f002:**
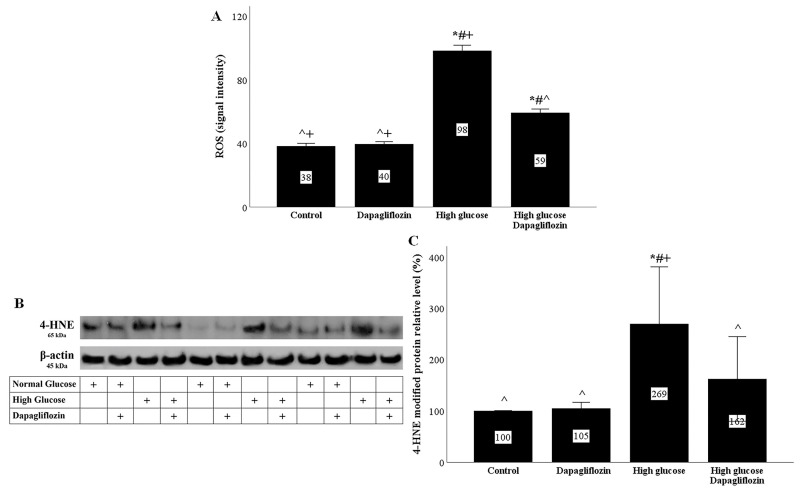
High glucose induced oxidative stress, whereas dapagliflozin ameliorated high-glucose-induced oxidative stress. High glucose increased ROS production, while dapagliflozin decreased high-glucose-induced ROS overproduction (**A**). High glucose enhanced the level of 4-HNE-modified proteins, whereas dapagliflozin reduced the level of high-glucose-induced 4-HNE-modified proteins (**B**,**C**). * *p* < 0.05 vs. control; # *p* < 0.05 vs. RPTECs treated with 15 ng/mL dapagliflozin; ^ *p* < 0.05 vs. RPTECs in high glucose; + *p* < 0.05 vs. RPTECs in high glucose and treated with 15 ng/mL dapagliflozin. 4-HNE, 4-Hydroxynonenal; ROS, reactive oxygen species.

**Figure 3 ijms-23-16107-f003:**
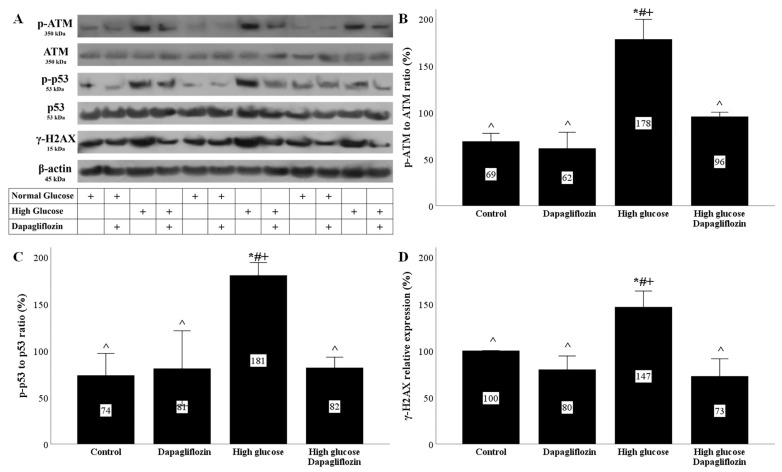
High glucose induced DNA damage and DNA damage response, but dapagliflozin prevented both. High glucose increased the p-ATM to ATM ratio, while dapagliflozin normalized it (**A**,**B**). High glucose upregulated the p-p53 to p53 ratio, but dapagliflozin prevented the high-glucose-induced increase in the p-p53 to p53 ratio (**A**,**C**). High glucose enhanced the level of γ-H2AΧ, whereas dapagliflozin normalized it (**A**,**D**). * *p* < 0.05 vs. control; # *p* < 0.05 vs. RPTECs treated with 15 ng/mL dapagliflozin; ^ *p* < 0.05 vs. RPTECs in high glucose; + *p* < 0.05 vs. RPTECs in high glucose and treated with 15 ng/mL dapagliflozin. γ-H2AX, phosphorylated at Ser139 histone H2AX; ATM, ataxia telangiectasia mutated kinase; p-ATM, phosphorylated at Ser1981 ATM; p53, tumor suppressor p53; p-p53, phosphorylated at Ser15 p53.

**Figure 4 ijms-23-16107-f004:**
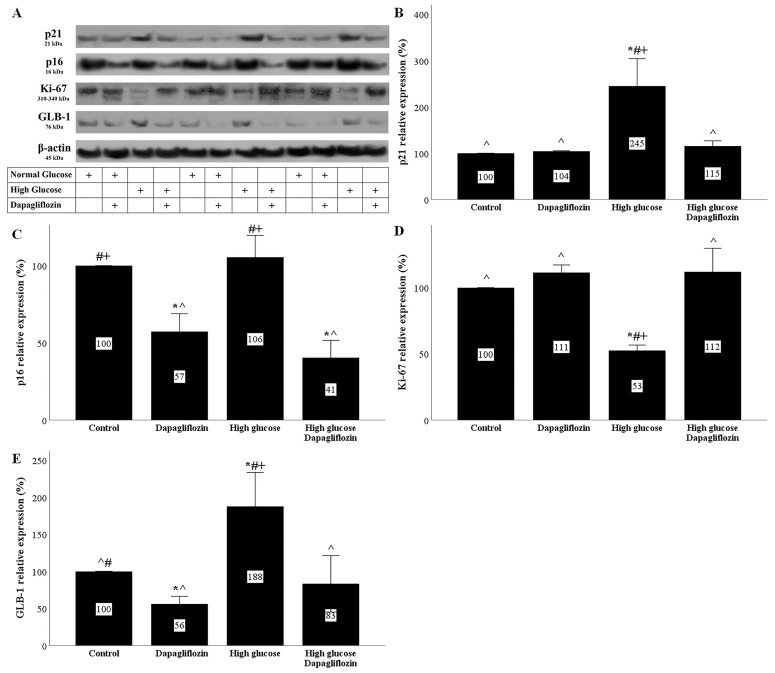
The effect of high glucose and dapagliflozin on cell cycle inhibitors p21 and p16, cell proliferation, and senescence. High glucose upregulated p21, while dapagliflozin normalized the p21 levels (**A**,**B**). Dapagliflozin downregulated p16 expression under normal and high-glucose conditions (**A**,**C**). High glucose decreased Ki-67, whereas dapagliflozin restored Ki-67 levels (**A**,**D**). High glucose increased GLB-1 expression, whereas dapagliflozin decreased GLB-1 under normal and high-glucose conditions (**A**,**E**). * *p* < 0.05 vs. control; # *p* < 0.05 vs. RPTECs treated with 15 ng/mL dapagliflozin; ^ *p* < 0.05 vs. RPTECs in high glucose; + *p* < 0.05 vs. RPTECs in high glucose and treated with 15 ng/mL dapagliflozin. GLB-1, beta-galactosidase; Ki-67, marker of proliferation Ki-67; p16, p16 INK4A, p21, p21 Waf1/Cip1.

**Figure 5 ijms-23-16107-f005:**
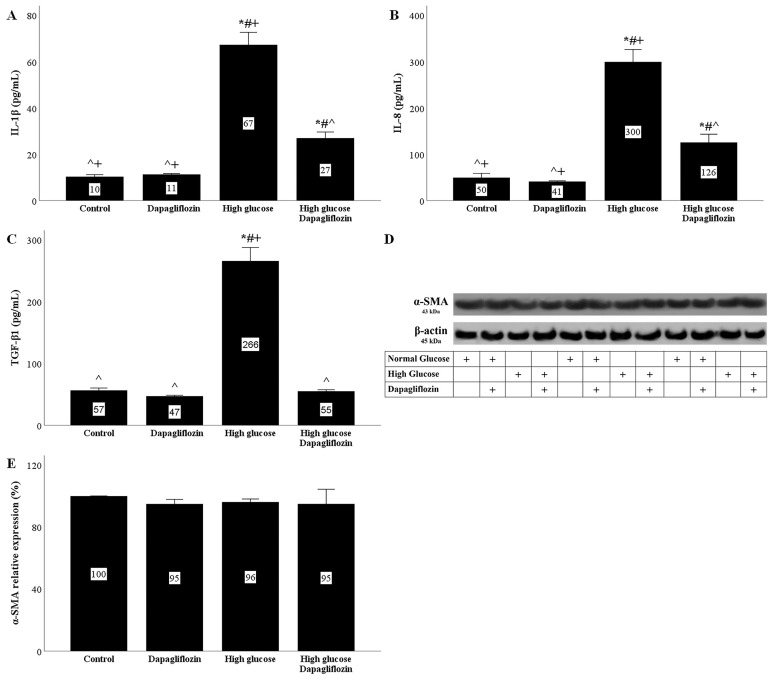
The effects of high glucose and dapagliflozin on the SASP phenotype and the epithelial-to-mesenchymal transition. High glucose triggered IL-1β production, but dapagliflozin decreased high-glucose-induced IL-1β overproduction (**A**). High glucose increased IL-8 production, whereas dapagliflozin reduced high-glucose-induced IL-8 overproduction (**B**). High glucose upregulated TGF-β1 production, but dapagliflozin normalized it (**C**). Neither high glucose nor dapagliflozin affected α-SMA (**D**,**E**). * *p* < 0.05 vs. control; # *p* < 0.05 vs. RPTECs treated with 15 ng/mL dapagliflozin; ^ *p* < 0.05 vs. RPTECs in high glucose; + *p* < 0.05 vs. RPTECs in high glucose and treated with 15 ng/mL dapagliflozin. α-SMA, α-smooth muscle actin; IL-1β, interleukin-1β; IL-8, interleukin-8; SASP, senescence-associated secretory phenotype; TGF-β1, transforming growth factor-β1.

**Figure 6 ijms-23-16107-f006:**
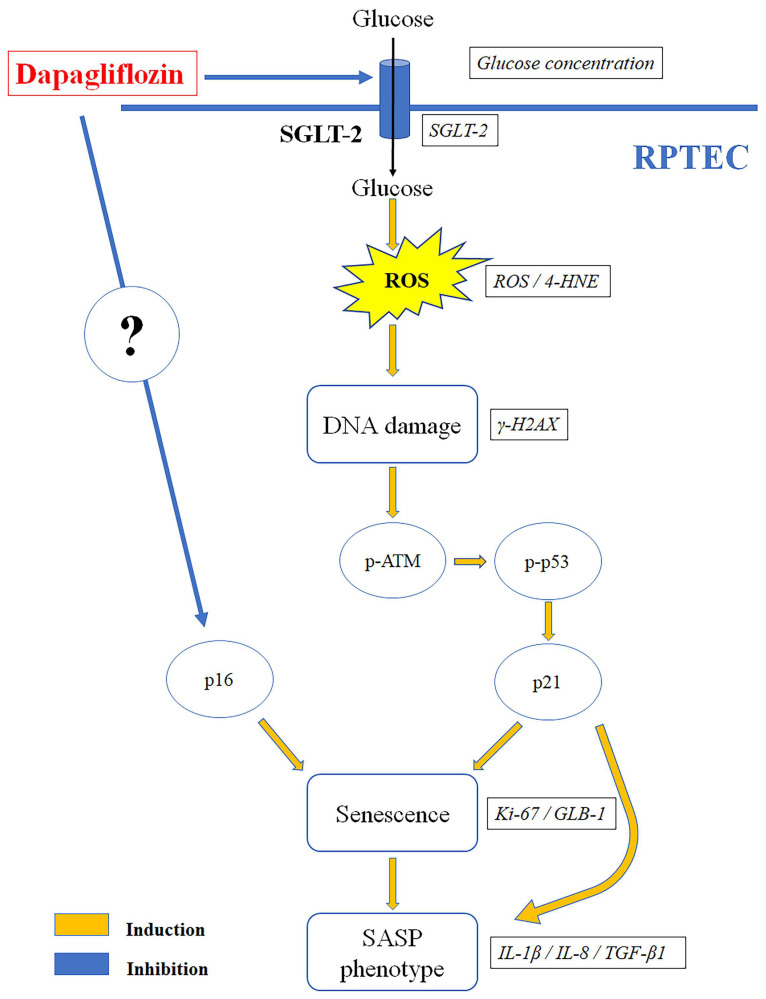
Dapagliflozin prevents high-glucose-induced RPTEC senescence. High glucose increased SGLT-2 expression and glucose consumption, enhancing ROS production. The latter induced DNA damage, ATM, and p53 phosphorylation. Stabilized p53 increased the cell cycle inhibitor p21, resulting in cell cycle arrest and increasing the cellular senescence marker GLB-1. In addition, RPTECs acquired am SASP phenotype, which was detected by the production of IL-1β, IL-8, and TGF-β1. By decreasing SGLT-2 expression and glucose consumption, dapagliflozin inhibited the above pathway and prevented RPTEC senescence. In addition, dapagliflozin reduced the cell cycle inhibitor p16 independently of the glucose conditions. 4-HNE, 4-Hydroxynonenal; γ-H2AX, phosphorylated at Ser139 histone H2AX; ATM, ataxia telangiectasia mutated kinase; GLB-1, beta-galactosidase; IL-1β, interleukin-1β; IL-8, interleukin-8; Ki-67, marker of proliferation Ki-67; p16, p16 INK4A, p21, p21 Waf1/Cip1; p53, tumor suppressor p53; ROS, reactive oxygen species; RPTEC, renal proximal tubular epithelial cell; SASP, senescence-associated secretory phenotype; SGLT-2, sodium–glucose cotransporter-2; TGF-β1, transforming growth factor-β1.

## Data Availability

The analyzed datasets generated during the study are available from the corresponding author upon reasonable request.

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
