# Peer review of "Dapagliflozin Prevents High-Glucose-Induced Cellular Senescence in Renal Tubular Epithelial Cells"

_ijms, 2022, doi:10.3390/ijms232416107_

Round 1

Reviewer 1 Report

The aim of this study was to demonstrate the effect of the drug dapagliflozin in preventing senescence of renal tubular epithelial cells induced by high glucose concentrations.

The study was designed and conducted correctly. The methodology used was appropriate based on the stated objective. The results were analyzed, described and presented clearly. The discussion is well substantiated based on previous studies. The conclusion is consistent with the results obtained.

However, it is necessary to made the minor corrections indicated in yellow in the attached file.

Author Response

We want to express our appreciation for the good comments. In the revised manuscript, we incorporated the suggested corrections.

Reviewer 2 Report

This paper is well written and above the average level of papers published in this journal. Readers will welcome the fact that the results are shown in a schematic diagram. I think this manuscript can be accepted as it is.

Author Response

Thank you for the very kind and encouraging comments.

Reviewer 3 Report

In this manuscript, the authors investigated the inhibitory effect of the antidiabetic drug dapagliflozin on high glucose-induced senescence in primary human renal tubular epithelial cells (RPTECs) and explored the possible molecular mechanisms. This work is very similar to the previously reported study by the same group of authors "A unifying model of glucotoxicity in human renal proximal tubular epithelial cells and the effect of the SGLT2 inhibitor dapagliflozin" published in 2020 (PMID: 32361978), which also explored the effect of dapagliflozin on RPTECs treated with high glucose, and also explored the changes of SCLT2 expression, ROS, TGF-β1 and IL-8. The authors should at least add some relevant animal experiments to show the novelty of this work. All the figures of this manuscript have really low quality. The manuscript was also poorly written.

1.      The method descriptions of many experiments are way too single.

2.      Only some Western blot experiments were carried out for the exploration of the molecular mechanism, and the Western blot bands are also of really low quality.

3.      In Figure 4D, does this treatment promote cell proliferation?

4.      The manuscript indicates that elevated ROS could lead to DNA damage, but further verification is needed, for example, after adding NAC, to verify whether the proteins associated with DNA damage are changed.

5.      In Figure 2A, ROS levels were significantly reduced after “high glucose+Dapagliflozin” treatment, and the authors believe that Dapagliflozin is acting through the ROS pathway. But why there is no difference between ROS levels in Dapagliflozin treated group and Control group? In addition, the ROS signal intensity was only 40-120? Is it 40-120%?

6.      In Figure 2B, why the 4-HNE protein expression levels of cells in the control group differed greatly between the first and second experiments. In addition, the difference in 4-HNE protein expression between the first and second experiments in the dapagliflozin group was also significant.

7.      Line 204-206 do not adequately explain the phenomenon described in line 202-203.

8.      Figures 5D and 5E show that high glucose treatment did not cause cellular fibrosis, which conflict with some of the relevant literature descriptions in the manuscript.

9.      Because the manuscript focuses on the oxidative damage to DNA caused by high glucose through ROS, it is more important to detect changes in the levels of DNA oxidative damage related indicators, such as 8-hydroxydeoxyguanosine (8-OHdG), than 4-HNE.

10.   For high glucose concentration, why chose 25 mM? The literature shows that 25 mM high glucose treatment for 24 h does not cause cellular senescence and ROS elevation. In addition, the high glucose concentration used in literature is mostly ≥50 mM.

11.   Many indicators in the control group vs. dapagliflozin group do not show difference, but there were some changes, such as P16 and GLB-1, what is the explanation for these differences?

12.   In Figure 6, the word italic is not used properly, "promote" and "inhibit" are not represented properly, and the Figure 6 is unnecessarily big.

13.   Does RPTEC in normal glucose plus dapagliflozin cause cellular glucose starvation?

14.   In line 313 of page 13, why using T-PER tissue protein extraction reagent for cellular experiments? It does not make any sense.

Author Response

Comment: The method descriptions of many experiments are way too single

Response: We provided adequate details for cell culture conditions and the WB assay. All the other measurements were performed with commercially available kits according to the manufacturer's instructions. We provided the names and the manufacturers of the kits in the manuscript, so anyone can see the details if they wish.

Comment: Only some Western blot experiments were carried out for the exploration of the molecular mechanism, and the Western blot bands are also of really low quality.

Response: WB assay is a handmade assay, and the WB bands provided with the manuscript are unmanipulated and of good quality. At least compared with the WB results of the most published papers.

Comment: In Figure 4D, does this treatment promote cell proliferation?

Response: No, it does not. The proper interpretation of this result is that high glucose decreases cell proliferation by inducing cellular senescence. This is in accordance with the effect of high glucose on p21. By preventing cellular senescence, dapaglifozin prevents high-glucose-induced downregulation of cell proliferation. Nowhere in the paper have we claimed something different than the above.

Comment: The manuscript indicates that elevated ROS could lead to DNA damage, but further verification is needed, for example, after adding NAC, to verify whether the proteins associated with DNA damage are changed.

Response: The fact that ROS induces DNA damage has been confirmed by too many papers. Thus, we were based on the established literature

Comment: In Figure 2A, ROS levels were significantly reduced after "high glucose+Dapagliflozin" treatment, and the authors believe that Dapagliflozin is acting through the ROS pathway. But why there is no difference between ROS levels in Dapagliflozin treated group and Control group? In addition, the ROS signal intensity was only 40-120? Is it 40-120%?

Response: We clearly stated and showed that dapagliflozin reduces ROS not directly but indirectly by decreasing glucose entry into the cells. The latter effect of dapagliflozin is significantly more potent under high glucose conditions. This is why its impact on ROS is observed under high glucose and not under normal glucose conditions. Signal intensity corresponds directly to the related units of measurement of the related kit.

Comment: In Figure 2B, why the 4-HNE protein expression levels of cells in the control group differed greatly between the first and second experiments. In addition, the difference in 4-HNE protein expression between the first and second experiments in the dapagliflozin group was also significant.

Response: This is common in WB assay, which is handmade. Besides, the three experiments were independent. Finally, WB is a semiquantitative assay showing the trend of protein level alteration under various conditions.

Comment: Line 204-206 do not adequately explain the phenomenon described in line 202-203.

Response: The same as in the response in the 5th comment.

Comment: Figures 5D and 5E show that high glucose treatment did not cause cellular fibrosis, which conflict with some of the relevant literature descriptions in the manuscript.

Response: The term cellular fibrosis does not exist. Fibrosis is always extracellular. Nowhere in the manuscript, we stated that high glucose does not induce fibrosis. The increase of TGF-β supports the opposite. We advocate that in our experimental model, high glucose does not cause epithelial to mesenchymal transition, as Figures 5D and 5E showed. In the discussion, we also stated that the increased TGF-β could induce fibrosis by affecting other cell types in the kidney.

Comment: Because the manuscript focuses on the oxidative damage to DNA caused by high glucose through ROS, it is more important to detect changes in the levels of DNA oxidative damage related indicators, such as 8-hydroxydeoxyguanosine (8-OHdG), than 4-HNE.

Response: We did not measure 4-HNE for showing DNA damage but as an addition to the measured ROS factor for supporting the effect of high glucose and dapagliflozin on oxidative stress. DNA damage was assessed with the γ-H2AX, an excellent indicator for this purpose, as well as with the p-ATM, and an indicator of DNA damage response.

Comment: For high glucose concentration, why chose 25 mM? The literature shows that 25 mM high glucose treatment for 24 h does not cause cellular senescence and ROS elevation. In addition, the high glucose concentration used in literature is mostly ≥50 mM.

Response: 25 mM is a pathophysiologically high glucose level since it corresponds to 450 mg/dL. In our opinion, a glucose concentration higher than 50 mM, i.e. >900mg/dL, is exceptionally toxic for cells mainly due to osmolarity reasons. Most papers use the concentration we use in this study. For instance, the very important studies by Brownlee and his team used a concentration very close to ours (30mM). 

Comment: Many indicators in the control group vs. dapagliflozin group do not show difference, but there were some changes, such as P16 and GLB-1, what is the explanation for these differences?

Response: We mentioned these differences in the manuscript and the need for further study. We also wrote some thoughts about the possible significance of p16 changes.

Comment: In Figure 6, the word italic is not used properly, "promote" and "inhibit" are not represented properly, and the Figure 6 is unnecessarily big.

Response: In italics are the factors that were experimentally assessed in the study. Induction and inhibition are correct. The figure's size can be changed during the manuscript's production by the journal.

Comment: Does RPTEC in normal glucose plus dapagliflozin cause cellular glucose starvation?

Response: Considering the initial concentration and the glucose consumption (shown in the related figure), at the end of the experiment and in the noted condition, there is still 45 mg/dL of glucose.

Comment: In line 313 of page 13, why using T-PER tissue protein extraction reagent for cellular experiments? It does not make any sense.

Response: This product is also widely used for protein extraction from cell cultures. We used this reagent for years in cell cultures, and it works well since it yields reasonable quantities of protein assessed with Bradford assay. Many other researchers use this reagent for cell cultures, for instance, of human breast cancer epithelial cell line (DOI: 10.18632/oncotarget.13013 / 10.3892/ijo.2018.4341), Human esophageal cancer cell line EC109 (DOI: 10.1186/1471-2199-13-4), Hepatocellular carcinoma cell lines (DOI: 10.18632/oncotarget.4455 / 10.1016/j.bbrc.2016.02.047 / 10.1038/s41598-019-45932-3), Lung cancer cell line (DOI: 10.3892/ol.2020.12309), human leukemia monocytic cell line and human embryonic kidney cell line (DOI: 10.1155/2019/4530534), colon cancer cell line (DOI: 10.1096/fj.201901255R), mouse neutrophils (DOI: 10.1007/s00441-021-03435-6), human myeloma cells (DOI: 10.1074/jbc.M508136200), peripheral blood mononuclear cells (DOI: 10.4049/jimmunol.174.10.6195), oral mucosal epithelial cells (DOI 10.1038/srep36266), mouse macrophages (DOI: 10.3390/ijms232213771) and many others.

For your convenience, I also present the related Question and Answer from the product FAQ at the manufacturer's site:

"I only want to buy one reagent for extraction of protein from both mammalian cells and tissues. Will M-PER Mammalian Protein Extraction Reagent work for protein extraction from tissue samples?

Answer

For extraction of protein from tissue samples as well as some difficult-to-lyse mammalian cells, we recommend using T-PER Tissue Protein Extraction Reagent (Cat. No. 78510) with homogenization."

Round 2

Reviewer 3 Report

Whatever